# Optimization of Ammonia Oxidation Using Response Surface Methodology

**Marek Inger \*****, Agnieszka Dobrzyńska-Inger, Jakub Rajewski** and **Marcin Wilk**

New Chemical Syntheses Institute, Al. Tysiąclecia Państwa Polskiego 13a, 24-100 Puławy, Poland;
agnieszka.dobrzynska-inger@ins.pulawy.pl (A.D.-I.); jakub.rajewski@ins.pulawy.pl (J.R.);
marcin.wilk@ins.pulawy.pl (M.W.)
**\*** Correspondence: marek.inger@ins.pulawy.pl; Tel.: +48-(81)-473-1415

**Abstract:** In this paper, the design of experiments and response surface methodology were proposed to study ammonia oxidation process. The following independent variables were selected: the reactor's load, the temperature of reaction and the number of catalytic gauzes, whereas ammonia oxidation efficiency and $N_2O$ concentration in nitrous gases were assumed as dependent variables (response). Based on the achieved results, statistically significant mathematical models were developed which describe the effect of independent variables on the analysed responses. In case of ammonia oxidation efficiency, its achieved value depends on the reactor's load and the number of catalytic gauzes, whereas the temperature in the studied range (870–910 °C) has no effect on this dependent variable. The concentration of nitrous oxide in nitrous gases depends on all three parameters. The developed models were used for the multi-criteria optimization with the application of desirability function. Sets of parameters were achieved for which optimization assumptions were met: maximization of ammonia oxidation efficiency and minimization of the $N_2O$ amount being formed in the reaction.

**Keywords:** ammonia oxidation; response surface methodology; desirability function; Box-Behnken design

## 1. Introduction

Nitric acid is mainly used for producing nitrogen fertilizers: ammonium nitrate (AN) and calcium ammonium nitrate (CAN) which constitute 75–80% of its entire production. The remaining amount of nitric acid is used in other industrial applications for example as a nitration agent for the production of explosives and other semi-organic products (aliphatic nitro compounds and aromatic nitro compounds) for the production of adipic acid, for metallurgy (etching steel) [1].

The industrial production of nitric acid is based on Ostwald process [1] which involves three basic stages: the catalytic oxidation of ammonia to nitrogen oxide (NO) with the use of oxygen from air, oxidation of nitrogen oxide (NO) to nitrogen dioxide ($NO_2$) and absorption of nitrogen oxides in water with the formation of $HNO_3$.

Ammonia consumption depends on the selectivity of the applied ammonia oxidation catalyst and on the process conditions. Among numerous catalysts [2–8], packages of gauzes made of noble metal alloys such as platinum and rhodium are most commonly applied in industrial practice [7–10]. Properly selected catalyst package allows to obtain ammonia conversion to main product (NO) in the range of 90–98% depending mainly on oxidation pressure [1,7,8]. Oxidation pressure has an inversely proportional effect on ammonia oxidation efficiency. In order to alleviate this effect, the temperature of reaction should be higher. However, this leads to the increased platinum losses and as a consequence, shortens the lifetime of the catalytic gauzes. For example, platinum losses are six times higher after

increasing the temperature of reaction from 820 to 920 °C [1,2]. Therefore, both these aspects should be taken into account to determine the temperature of reaction.

The application of medium pressure in the oxidation unit (0.35–0.55 MPa) and high pressure in the absorption unit (0.8–1.5 MPa) is optimal for specific ammonia consumption and efficient energy use. Therefore, modern nitric acid plants are dual-pressure ones. The average pressure in the oxidation unit is a kind of trade-off between the capacity that is possible to achieve per 1 $m^2$ of catalytic gauzes, oxidation efficiency, number of gauzes in package, lifetime of gauzes and noble metals losses during exploitation [1,2].

In the context of global warming and climate changes, a very important issue related to ammonia oxidation process is the amount of the by-product formed that is nitrous oxide ($N_2O$). In Kyoto Protocol, $N_2O$ was qualified as a greenhouse gas with a very high global warming potential, about 300 times higher than $CO_2$ [11]. At room temperature, $N_2O$ is a colourless, non-flammable gas with a delicate pleasant smell and sweet taste [12]. Since it was isolated at the end of 17th century and because of its pain-relieving and anaesthetic properties, it has been widely applied in dentistry and surgery. Currently, due to some concerns, there is an ongoing discussion on its safe use which has the effect of decreasing the $N_2O$ application in medicine [12,13]. At the same time, the increasing trend of its use for recreational purposes is observed. Inhaling the 'laughing gas' causes euphoria and hallucinations [13].

Microbial nitrification and de-nitrification in land and aqueous eco-systems are the natural sources of $N_2O$ in environment. The anthropogenic sources are cultivated soils fertilized intensely with nitrogen fertilizers and industrial processes such as burning fossil fuels and biomass as well as the production of adipic acid and nitric acid with the last one being regarded as the biggest source of $N_2O$ in the chemical industry [14,15]. Nitrous oxide formed in nitric acid plant does not undergo any conversions and it is released to atmosphere. Currently, the emission of this gas is monitored and industry is obliged to reduce it. Pursuant to BAT requirements, concentration of this gas in outlet gases cannot exceed 20–300 ppm depending on the type of nitric acid plants [16,17]. However, due to the battle against climate change and global warming, further restrictions in emission limits can be expected.

There are several methods of limiting $N_2O$ emissions from nitric acid plants [14]. Generally, they can be classified as primary and secondary methods. Primary methods involve preventing the formation of $N_2O$ during ammonia oxidation. They include modification of catalytic gauzes (so-called low-emissions systems) and parameters optimization of ammonia oxidation process. Secondary methods involve the removal of $N_2O$. At the temperature over 800 °C, thermal decomposition of $N_2O$ occurs but the efficient decomposition requires ensuring adequately long residence time at high temperatures [14,15].

The achievement of low level of $N_2O$ emissions requires the application of the catalytic methods such as high temperature $N_2O$ decomposition from nitrous gases, low- or middle temperature $N_2O$ decomposition or reduction from tail gases. High temperature method is more common. In some cases, the combination of primary method (application of modified catalytic gauzes packages and/or optimization of ammonia oxidation parameters) and high temperature $N_2O$ decomposition ensures meeting the emission standards.

Optimization of production process requires extensive knowledge and understanding the effect of particular parameters on the process. Until recently, the most commonly applied approach of researchers to study simple and complex processes was 'one-factor-at-a-time' (*OFAT*), which is time consuming and ineffective method for processes with multiple complicated dependencies between parameters. Over the last years, mathematical and statistical methods for design of experiments and parameters optimization have been applied more frequently [18]. Because of its usability, this method is applied for the design, improvement and optimization of production processes and products [19–21]. It is a widely applied method in research of various processes [22–27] and approx. 50% of all applications is in medicine, engineering, biochemistry, physics and computer science [28]. In this

method, reaction kinetics equations and process mechanism are not taken into account and they are regarded as a 'black-box' [29] (Figure 1).

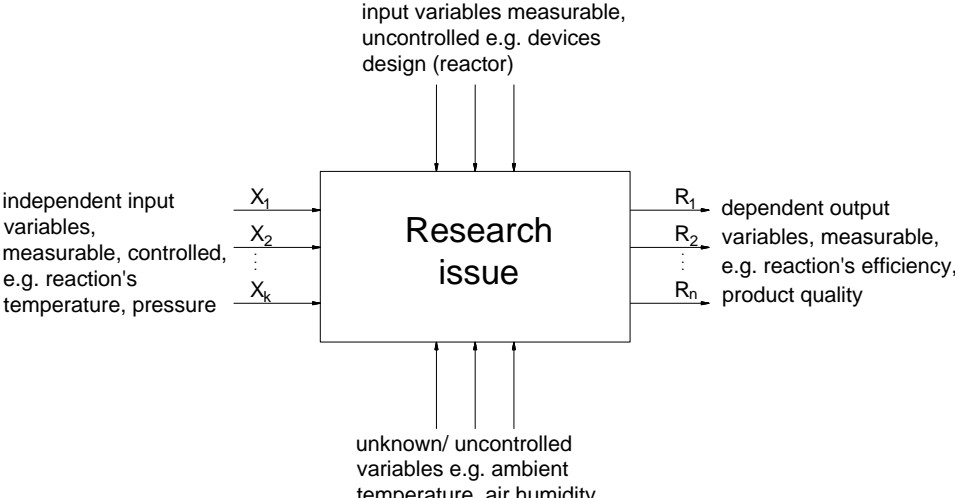

**Figure 1.** "Black–box" model of the research issue in design of experiments methodology.

The choice of experiment plan depends mainly on the issue which is the subject matter of investigations as well as on objectives which are set. The most commonly applied experiments plans include: full or fractional factorial, Plackett-Burman, central composite, Box-Behnken and Taguchi designs.

As a result of modelling of the data obtained, empirical equations with statistically significant importance are received which describe the effect of process variables (independent variables) on the process result (response variable).

Desirability function (*DF*) can be applied in search for optimal operational parameters. The method proposed by Derringer and Such [18] involves the construction response surface model and then finding the values of independent variables which ensure the most desirable value. The objective of the presented studies was the analysis of the impact of reactor's operational parameters on ammonia oxidation reaction. To the best of our knowledge, the approach presented here to describe ammonia oxidation process is published for the first time.

## 2. Results and Discussion

### 2.1. Design of Experiments

Ammonia oxidation reaction depends on a few process variables. In this study, the effect of the reactor's load ($X_1$), the temperature of nitrous gas specifying the temperature of reaction ($X_2$) and the number of catalytic gauzes ($X_3$) on ammonia oxidation reaction was investigated. The oxidation efficiency of $NH_3$ to $NO$ ($R_1$) and $N_2O$ concentration in nitrous gases ($R_2$) were selected as measures for ammonia oxidation reaction. The matrix of 15 experiments including particular levels of coded variables and achieved values of response variables $R_1$ and $R_2$ are presented in Table 1. In the regarded experimental area of independent variables, ammonia oxidation efficiency ranged from 91.4% to 96.4%, whereas $N_2O$ concentration in nitrous gases ranged from 1011 to 1762 ppm.

**Table 1.** The Box-Behnken design matrix and experimental data.

| Standard Order | Run Order | $X_1$ Reactor's Load, kg $NH_3$/($m^2$h) | $X_2$ Temperature, °C | $X_3$ No. of Gauzes, pcs | $R_1$ Ammonia Oxidation Efficiency, % | $R_2$ $N_2O$ Concentration in Nitrous Gases, ppm |
|---|---|---|---|---|---|---|
| 3 | 1 | −1 | 1 | 0 | 96.1 | 1011 |
| 15 | 2 | 0 | 0 | 0 | 96.1 | 1279 |
| 7 | 3 | −1 | 0 | 1 | 96.2 | 1238 |
| 6 | 4 | 1 | 0 | −1 | 91.4 | 1620 |
| 9 | 5 | 0 | −1 | −1 | 92.0 | 1762 |
| 5 | 6 | −1 | 0 | −1 | 93.7 | 1348 |
| 14 | 7 | 0 | 0 | 0 | 96.2 | 1265 |
| 12 | 8 | 0 | 1 | 1 | 96.4 | 1074 |
| 2 | 9 | 1 | −1 | 0 | 95.6 | 1457 |
| 8 | 10 | 1 | 0 | 1 | 96.2 | 1312 |
| 1 | 11 | −1 | −1 | 0 | 96.3 | 1423 |
| 4 | 12 | 1 | 1 | 0 | 95.8 | 1114 |
| 13 | 13 | 0 | 0 | 0 | 96.2 | 1271 |
| 11 | 14 | 0 | −1 | 1 | 96.0 | 1506 |
| 10 | 15 | 0 | 1 | −1 | 92.7 | 1207 |

## 2.2. Model Fitting

The first task was to find out which equation would allow to obtain the best correlation between independent variables and responses. Analysis of Variance (ANOVA) was carried out for most frequently applied equations: linear, two-factor interaction (2FI), quadratic and cubic. Table 2 includes the summary statistics of both responses for different mathematical equations.

**Table 2.** Model summary statistics for response variables $R_1$ and $R_2$.

| Response Variable: $R_1$—Ammonia Oxidation Efficiency | | | | | | |
|---|---|---|---|---|---|---|
| Source | Std. Dev. | $R^2$ | Adjusted $R^2$ | Predicted $R^2$ | PRESS | |
| Linear | 1.08 | 0.6969 | 0.6142 | 0.4268 | 24.38 | |
| 2FI | 1.20 | 0.7294 | 0.5265 | −0.1215 | 47.70 | |
| Quadratic | 0.2790 | 0.9908 | 0.9744 | 0.8557 | 6.14 | Suggested |
| Cubic | 0.0577 | 0.9998 | 0.9989 | | * | Aliased |
| Response variable: $R_2$—$N_2O$ concentration in nitrous gases | | | | | | |
| Source | Std. Dev. | $R^2$ | Adjusted $R^2$ | Predicted $R^2$ | PRESS | |
| Linear | 87.82 | 0.8524 | 0.8121 | 0.6993 | $1.728 \times 10^5$ | |
| 2FI | 93.58 | 0.8781 | 0.7867 | 0.4155 | $3.359 \times 10^5$ | |
| Quadratic | 49.78 | 0.9784 | 0.9396 | 0.6574 | $1.969 \times 10^5$ | Suggested |
| Cubic | 7.02 | 0.9998 | 0.9988 | | * | Aliased |

* - case(s) with leverage of 1.0000; PRESS statistic not defined.

Based on the achieved results, it was found that the experimental data is described best with quadratic and cubic equations. For both responses, high values of $R^2$ and adjusted $R^2$ were achieved. The number of conducted experiments caused that the cubic model was aliased. It means that the experimental matrix contains an insufficient number of experimental points for independent estimation of all effects for these models. Therefore, quadratic equation was selected for further analysis.

The statistical significance of these equations and their particular terms was specified based on Analysis of Variance (ANOVA). Results of this analysis are presented in Tables 3 and 4, for response variables $R_1$ and $R_2$ respectively. Large $F$-value indicates that most changes of independent variable can be explained with the developed regression equation. The correlated probability $p$-value is used to estimate whether $F$-value is large enough to show statistical significance.

**Table 3.** ANOVA results for response variable $R_1$.

| Source | Sum of Squares | df | Mean Square | $F$-Value | $p$-Value | Significance |
|---|---|---|---|---|---|---|
| Model | 42.14 | 9 | 4.68 | 60.16 | 0.0001 | highly significant |
| $X_1$ | 1.36 | 1 | 1.36 | 17.49 | 0.0086 | significant |
| $X_2$ | 0.1512 | 1 | 0.1512 | 1.94 | 0.2221 | not significant |
| $X_3$ | 28.13 | 1 | 28.13 | 361.35 | <0.0001 | highly significant |
| $X_1X_2$ | 0.0400 | 1 | 0.0400 | 0.5139 | 0.5055 | not significant |
| $X_1X_3$ | 1.32 | 1 | 1.32 | 16.99 | 0.0092 | significant |
| $X_2X_3$ | 0.0225 | 1 | 0.0225 | 0.2891 | 0.6139 | not significant |
| $X_1{}^2$ | 0.0126 | 1 | 0.0126 | 0.1614 | 0.7044 | not significant |
| $X_2{}^2$ | 0.0926 | 1 | 0.0926 | 1.19 | 0.3252 | not significant |
| $X_3{}^2$ | 11.09 | 1 | 11.09 | 142.53 | < 0.0001 | highly significant |
| Residual | 0.3892 | 5 | 0.0778 | | | |
| Lack of Fit | 0.3825 | 3 | 0.1275 | 38.25 | 0.0256 | significant |
| Pure Error | 0.0067 | 2 | 0.0033 | | | |
| Corrected total SS | 42.53 | 14 | | | | |

$p < 0.0001$—highly significant, $0.0001 < p < 0.05$—significant, $p > 0.05$—not significant

The probability $p$-value for the achieved model of variable $R_1$ is 0.0001. It means that the model is statistically significant but some terms of equation are statistically not significant. Coefficients: $R^2$, adjusted $R^2$ and predicted $R^2$ are very high: 0.9908, 0.9744 and 0.8577, respectively. There is also high compliance between coefficients: predicted $R^2$ and adjusted $R^2$ (difference <0.2). The achievement of statistically significant value lack of fit (0.0256) is the incompliance of this model as this parameter should be statistically not significant.

**Table 4.** ANOVA results for response variable $R_2$.

| Source | Sum of Squares | df | Mean Square | $F$-Value | $p$-Value | Significance |
|---|---|---|---|---|---|---|
| Model | $5.623 \times 10^5$ | 9 | 62,480.28 | 25.21 | 0.0012 | highly significant |
| $X_1$ | 29,161.12 | 1 | 29,161.12 | 11.77 | 0.0186 | significant |
| $X_2$ | $3.793 \times 10^5$ | 1 | $3.793 \times 10^5$ | 153.05 | <0.0001 | highly significant |
| $X_3$ | 81,406.13 | 1 | 81,406.13 | 32.85 | 0.0023 | significant |
| $X_1X_2$ | 1190.25 | 1 | 1190.25 | 0.4803 | 0.5192 | not significant |
| $X_1X_3$ | 9801.00 | 1 | 9801.00 | 3.95 | 0.1034 | not significant |
| $X_2X_3$ | 3782.25 | 1 | 3782.25 | 1.53 | 0.2716 | not significant |
| $X_1{}^2$ | 732.33 | 1 | 732.33 | 0.2955 | 0.6101 | not significant |
| $X_2{}^2$ | 148.10 | 1 | 148.10 | 0.0598 | 0.8166 | not significant |
| $X_3{}^2$ | 54,881.26 | 1 | 54,881.26 | 22.14 | 0.0053 | significant |
| Residual | 12,391.92 | 5 | 2478.38 | | | |
| Lack of Fit | 12,293.25 | 3 | 4097.75 | 83.06 | 0.0119 | significant |
| Pure Error | 98.67 | 2 | 49.33 | | | |
| Corrected total SS | $5.7471 0^5$ | 14 | | | | |

$p < 0.0001$—highly significant, $0.0001 < p < 0.05$—significant, $p > 0.05$—not significant

In case of response variable $R_2$, the probability $p$-value (0.0012) indicates that the assumed quadratic equation is statistically significant but some of its terms are statistically not significant. High coefficients $R^2$, adjusted $R^2$ and predicted $R^2$ are also achieved for the second response variable and they are: 0.9784, 0.9396 and 0.6574, respectively. However, the difference between predicted $R^2$ and adjusted $R^2$ is larger than the recommended one (>0.2). This may demonstrate a large block effect or problems with model or data. This model is also characteristic of statistically significant parameter lack of fit ($p = 0.0119$).

At a further stage of analysis, statistically not significant terms of initial equation were eliminated from the analysis. The reduction was made using step-by-step method (from the most insignificant term). For both these response variables, only statistically significant terms were left and higher $R^2$,

adjusted $R^2$ and predicted $R^2$ coefficients were achieved. Results of Analysis of Variance (ANOVA) are presented in Tables 5 and 6.

**Table 5.** ANOVA results for reduced model of the response variable $R_1$.

| Source | Sum of Squares | df | Mean Square | *F*-Value | *p*-Value | Significance |
|---|---|---|---|---|---|---|
| Model | 41.83 | 4 | 10.46 | 148.66 | <0.0001 | highly significant |
| $X_1$ | 1.36 | 1 | 1.36 | 19.35 | 0.0013 | significant |
| $X_3$ | 28.13 | 1 | 28.13 | 399.85 | <0.0001 | highly significant |
| $X_1X_3$ | 1.32 | 1 | 1.32 | 18.80 | 0.0015 | significant |
| $X_3{}^2$ | 11.02 | 1 | 11.02 | 156.63 | <0.0001 | highly significant |
| Residual | 0.7034 | 10 | 0.0703 | | | |
| Lack of Fit | 0.6967 | 8 | 0.0871 | 26.13 | 0.0374 | significant |
| Pure Error | 0.0067 | 2 | 0.0033 | | | |
| Corrected total SS | 42.53 | 14 | | | | |
| $R^2$ | 0.9835 | | | | | |
| Adjusted $R^2$ | 0.9768 | | | | | |
| Predicted $R^2$ | 0.9550 | | | | | |

The obtained mathematical model for response $R_1$ is highly significant (*p*-value < 0.0001). The dependence on linear terms $X_1$, $X_3$, interaction $X_1X_3$ and quadratic term $X_3{}^2$ are significant. High determination coefficients are obtained l ($R^2$ = 0.9835, adjusted $R^2$ = 0.9768, predicted $R^2$ = 0.9550). The final model is presented in Equation (1).

$$R \ = \ 96.04 - 0.4125X_1 \ + \ 1.88X_3 \ + \ 0.575X_1X_3 - 1.72X_3^2 \tag{1}$$

**Table 6.** ANOVA results for reduced model of the response variable $R_2$.

| Source | Sum of Squares | df | Mean Square | *F*-Value | *p*-Value | Significance |
|---|---|---|---|---|---|---|
| Model | $5.467 \times 10^5$ | 4 | $1.367 \times 10^5$ | 48.81 | <0.0001 | highly significant |
| $X_1$ | 29,161.12 | 1 | 29,161.12 | 10.41 | 0.0091 | significant |
| $X_2$ | $3.793 \times 10^5$ | 1 | $3.793 \times 10^5$ | 135.47 | <0.0001 | highly significant |
| $X_3$ | 81,406.13 | 1 | 81,406.13 | 29.07 | 0.0003 | significant |
| $X_3{}^2$ | 56,826.52 | 1 | 56,826.52 | 20.30 | 0.0011 | significant |
| Residual | 28,000.13 | 10 | 2800.01 | | | |
| Lack of Fit | 27,901.46 | 8 | 3487.68 | 70.70 | 0.0140 | significant |
| Pure Error | 98.67 | 2 | 49.33 | | | |
| Corrected total SS | $5.747 \times 10^5$ | 14 | | | | |
| $R^2$ | 0.9513 | | | | | |
| Adjusted $R^2$ | 0.9318 | | | | | |
| Predicted $R^2$ | 0.8743 | | | | | |

The obtained mathematical model for response $R_2$ is highly significant (*p*-value < 0.0001). The dependence on linear terms $X_1$, $X_2$, $X_3$ and quadratic term $X_3{}^2$ are significant. High determination coefficients ($R^2$ = 0.9513, adjusted $R^2$ = 0.9318, predicted $R^2$ = 0.8743) were achieved for the model. The final model is presented in Equation (2).

$$R \ = \ 1260 \ + \ 60.37X_1 - 217.75X_2 - 100.88X_3 \ + \ 123.37X_3^2 \tag{2}$$

*2.3. Model Diagnostics*

Before the process optimization, the model diagnostics for both equations was performed because of occurrence of statistically significant *Lack of fit* parameter. Results of model diagnostics: a normal probability of the residuals, residuals analysis and actual data versus predicted values plots were analysed.

Figure 2 presents model diagnostics for response variable $R_1$, whereas Figure 3 presents model diagnostics for response variable $R_2$. Normal plot of studentised residuals should be approximately a straight line, whereas studentised residuals versus predicted response values and versus run should be a random scatter. Points in plots of real response values with reference to predicted response values line up accurately along the axis at the angle of 45°.

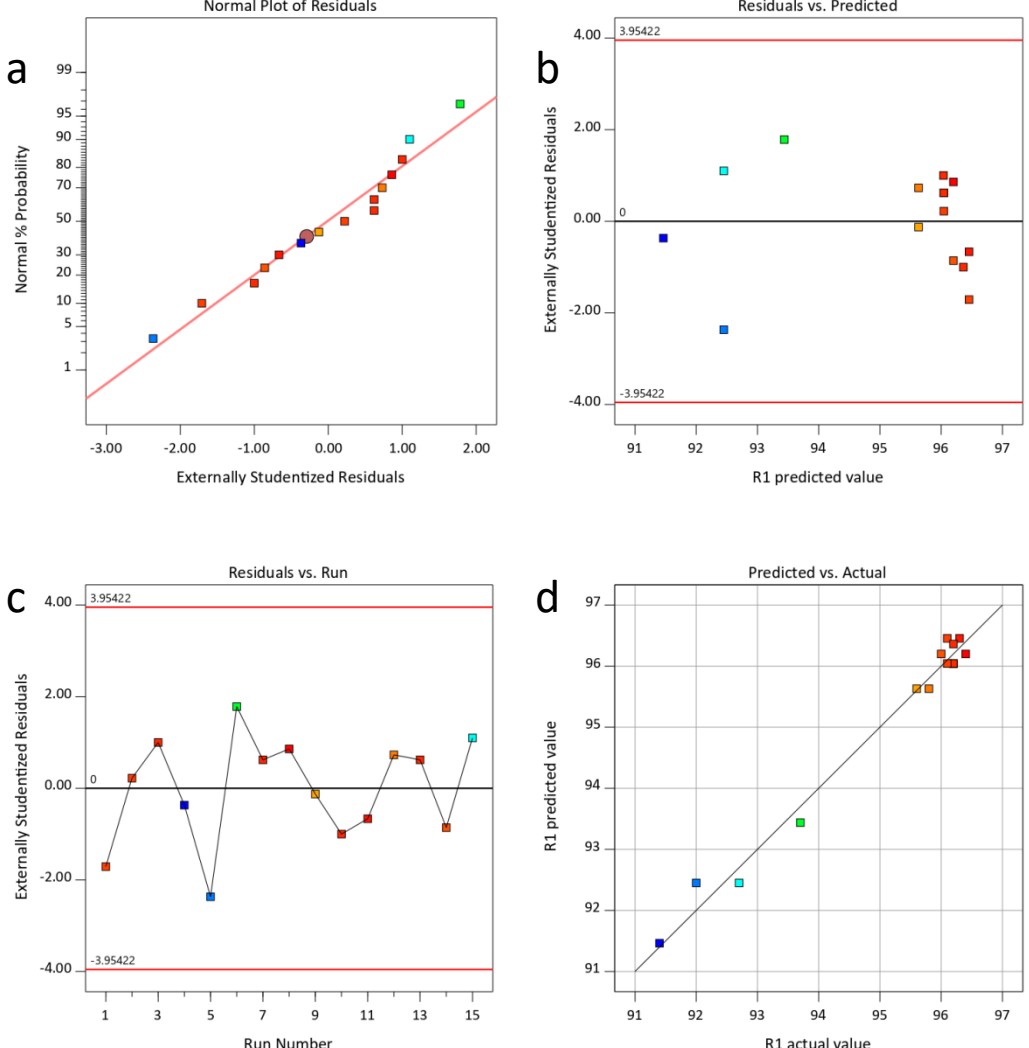

**Figure 2.** Model diagnostics for response variable $R_1$ (**a**) A normal probability plot of the residuals; (**b**) Residuals versus predicted value of $R_1$; (**c**) Residuals versus run number; (**d**) Predicted versus actual value of response variable $R_1$. Points on graphs correspond to results of particular experiments and colour points correspond to the value in accordance with the scale.

These diagnostics show that despite the fact that *Lack of fit* parameter is statistically significant, experimental and predicted points for both equations correlate well with each other.

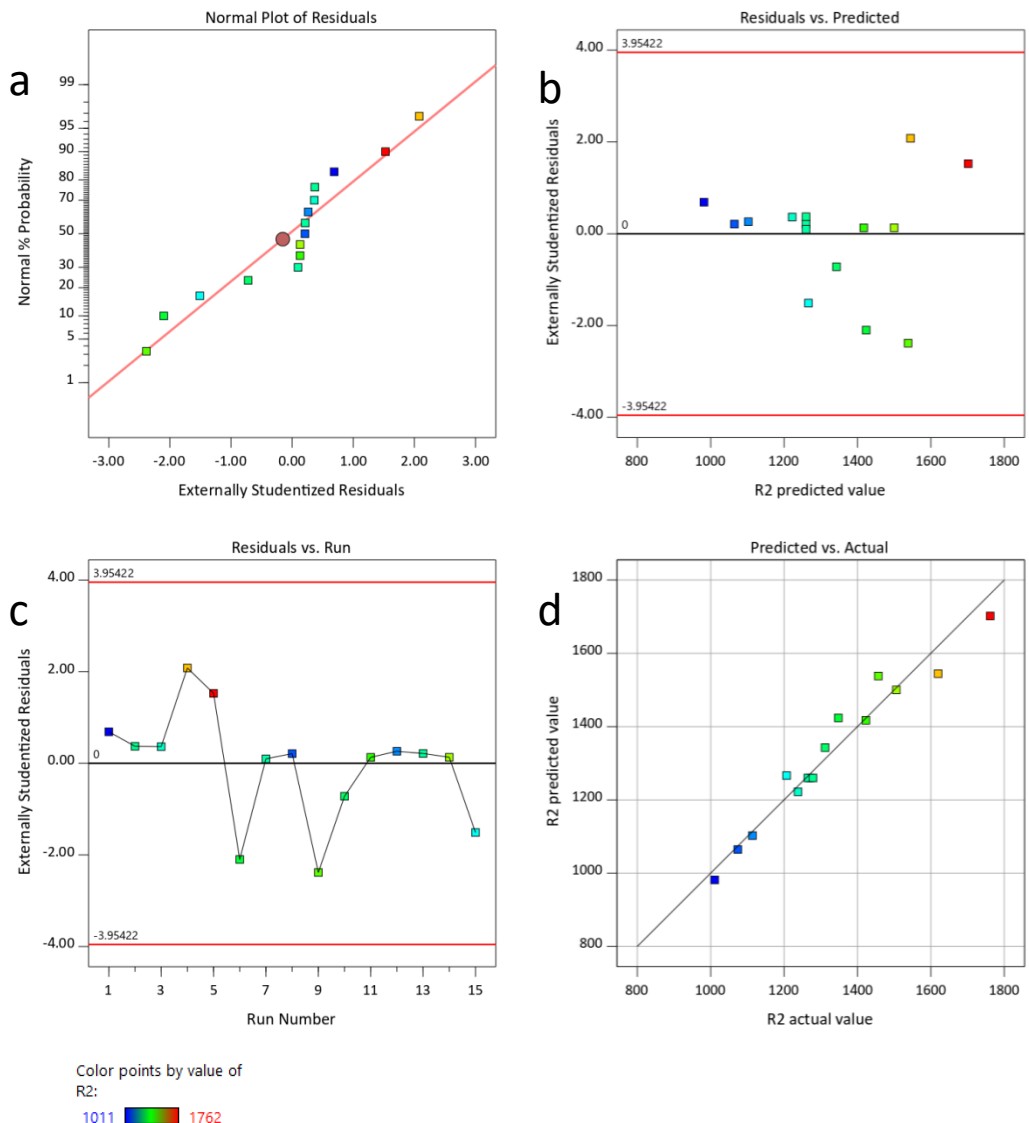

**Figure 3.** Model diagnostics for response variable $R_2$ (**a**) A normal probability plot of the residuals; (**b**) Residuals versus predicted value of $R_2$; (**c**) Residuals versus run number; (**d**) Predicted versus actual value of response variable $R_2$. Points on graphs correspond to results of particular experiments and colour points correspond to the value in accordance with the scale.

## 2.4. The Effect of Independent Variables

In case of response variable $R_1$ (ammonia oxidation efficiency), the mathematical model shows a strong linear effect of the reactor's load ($X_1$) and the number of catalytic gauzes ($X_3$) and interaction between these two variables ($X_1X_3$) and the quadratic term number of catalytic gauzes ($X_3^2$) on the achieved response variable. The temperature of reaction in the studied range does not affect the ammonia oxidation efficiency. The effect of $X_1$ and $X_3$ on response $R_1$ were shown as contour plot (Figure 4). According to the presented plot of variable of $R_1$, a small number of catalytic gauzes causes lower ammonia oxidation efficiency for the entire range of studies reactor's load. The increase in the number of catalytic gauzes to $X_3 = 0$ causes increase in oxidation efficiency within the entire range of studied reactor's load.

Studies related to dependency of $N_2O$ concentration in nitrous gases on operating parameters are relatively new research issue. Therefore, there is a lack of scientific reports dedicated to systematic studies in this field. In case of the $N_2O$ concentration in nitrous gases, the achieved mathematical

model demonstrates a significant effect of the selected process variables ($X_1$, $X_2$, $X_3$) and the quadratic term number of catalytic gauzes ($X_3{}^2$) on the achieved response variable. This effect is illustrated in Figures 5–7. The analysis of Equation (2) and Figures 5–7 indicates that the temperature of reaction has the biggest quantitative effect on $N_2O$ concentration in nitrous gases. From the comparison of plots (Figure 5a–c) it can be concluded that despite the presence of statistically significant terms of equations derived from variable $X_3$, plots of contour line corresponding to levels 0 and 1 are similar. Only for $X_3 = -1$, higher values of $R_2$ are achieved. Profiles of response variable $R_2$ presented in Figures 6a–c and 7a–c confirm the effect of the number of catalytic gauzes. Both these figures show that the number of catalytic gauzes has little effect on the amount of $N_2O$ being formed. For the level of $X_3 = 0$–0.4 (10–12 gauzes), the local optimum is observed. For this number of gauzes, increasing the reactor's load ($X_1 = 1$) at the fixed reaction temperature (Figure 6a–c) and decreasing the reaction temperature at the fixed reactor's load (Figure 7a–c) does not cause a significant decrease in $N_2O$ concentration in nitrous gases.

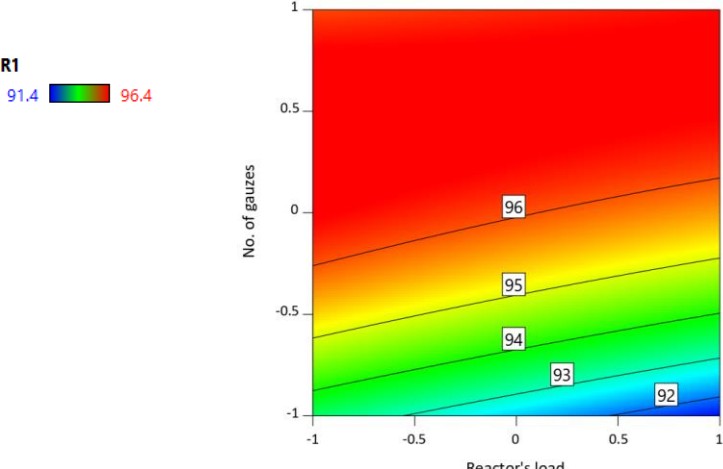

**Figure 4.** Interaction effect between the reactor's load ($X_1$) and the number of catalytic gauzes ($X_3$) on ammonia oxidation efficiency ($R_1$) by contour plot.

## 2.5. Multi-Response Desirability Optimization

The major optimization task is to find the number of catalytic gauzes and the permissible reactor's load ensuring the maximization of ammonia oxidation efficiency ($R_1$) and the minimization of $N_2O$ concentration in nitrous gases. Results of experiments discussed in Section 2.4. indicate that statistically, the temperature has no significant effect on ammonia oxidation efficiency but on the other hand, the amount of $N_2O$ formed is reversely proportional to the temperature of reaction. For desirability function, it was assumed that independent variables are in the variability range. Assumptions for the optimization are presented in Table 7.

**Table 7.** Assumptions for the optimization of the ammonia oxidation process using desirability function. Variables symbol identification according to the Table 1.

| Name | Goal | Lower Limit | Upper Limit | Lower Weight * | Upper Weight * | Importance ** |
|------|------|-------------|-------------|----------------|----------------|---------------|
| $X_1$ | in range | −1 | 1 | 1 | 1 | 3 |
| $X_2$ | in range | −1 | 1 | 1 | 1 | 3 |
| $X_3$ | in range | −1 | 1 | 1 | 1 | 3 |
| $R_1$ | maximize | 91.4 | 96.4 | 1 | 1 | 3 |
| $R_2$ | minimize | 1011 | 1762 | 1 | 1 | 3 |

* Weight: 1—linear change of values in the range from 0 to 1; ** Importance: 5—high, 3—medium, 1—low.

For such optimization assumptions, the area of detailed set of parameters was achieved. It confirms that optimization assumptions are met. Desirability functions for three temperature levels are presented as contour plot in Figure 8.

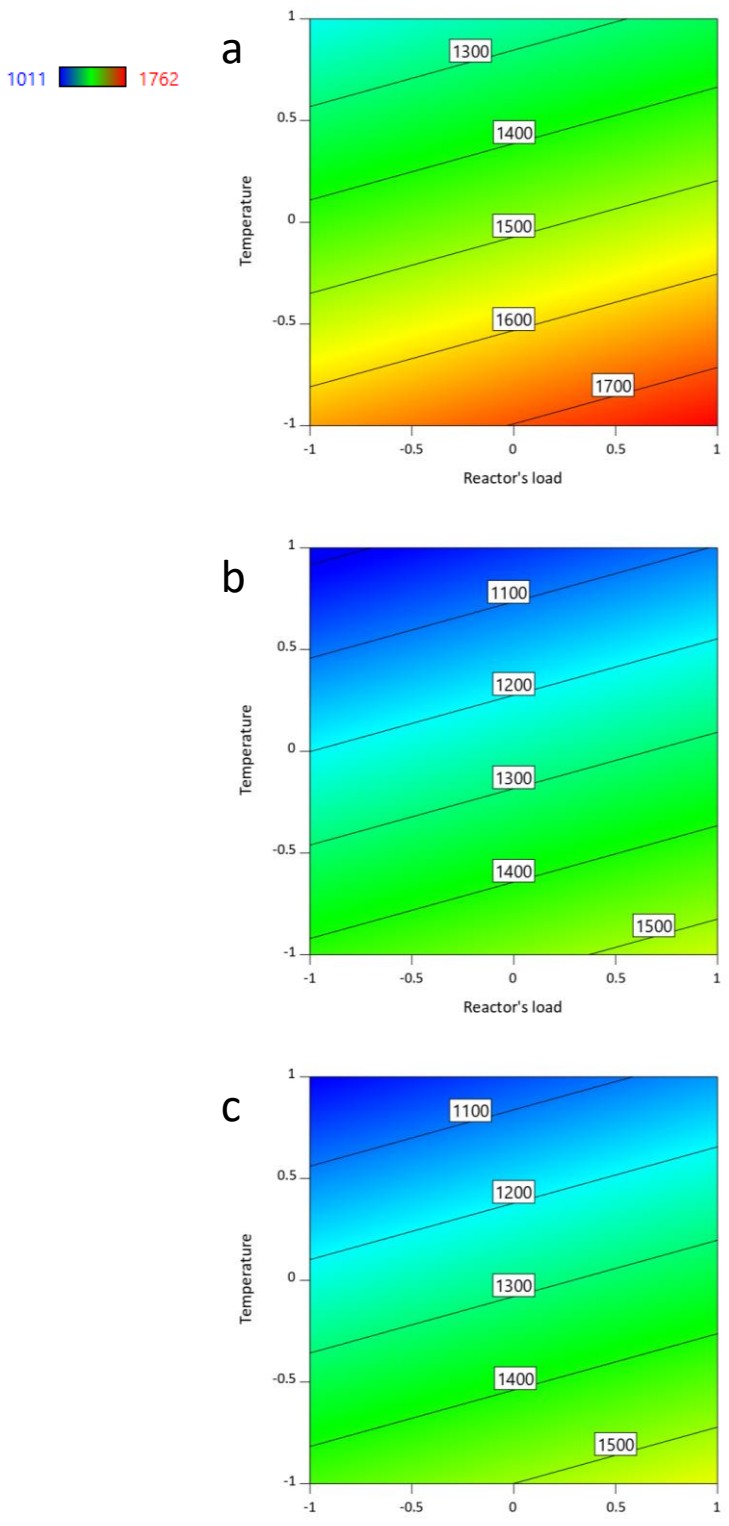

**Figure 5.** Interaction effect between the reactor's load ($X_1$) and the temperature ($X_2$) at fixed number of catalytic gauzes ($X_3$) on $N_2O$ concentration in nitrous gases ($R_2$) by contour plot. (**a**) $X_3 = -1$; (**b**) $X_3 = 0$; (**c**) $X_3 = 1$.

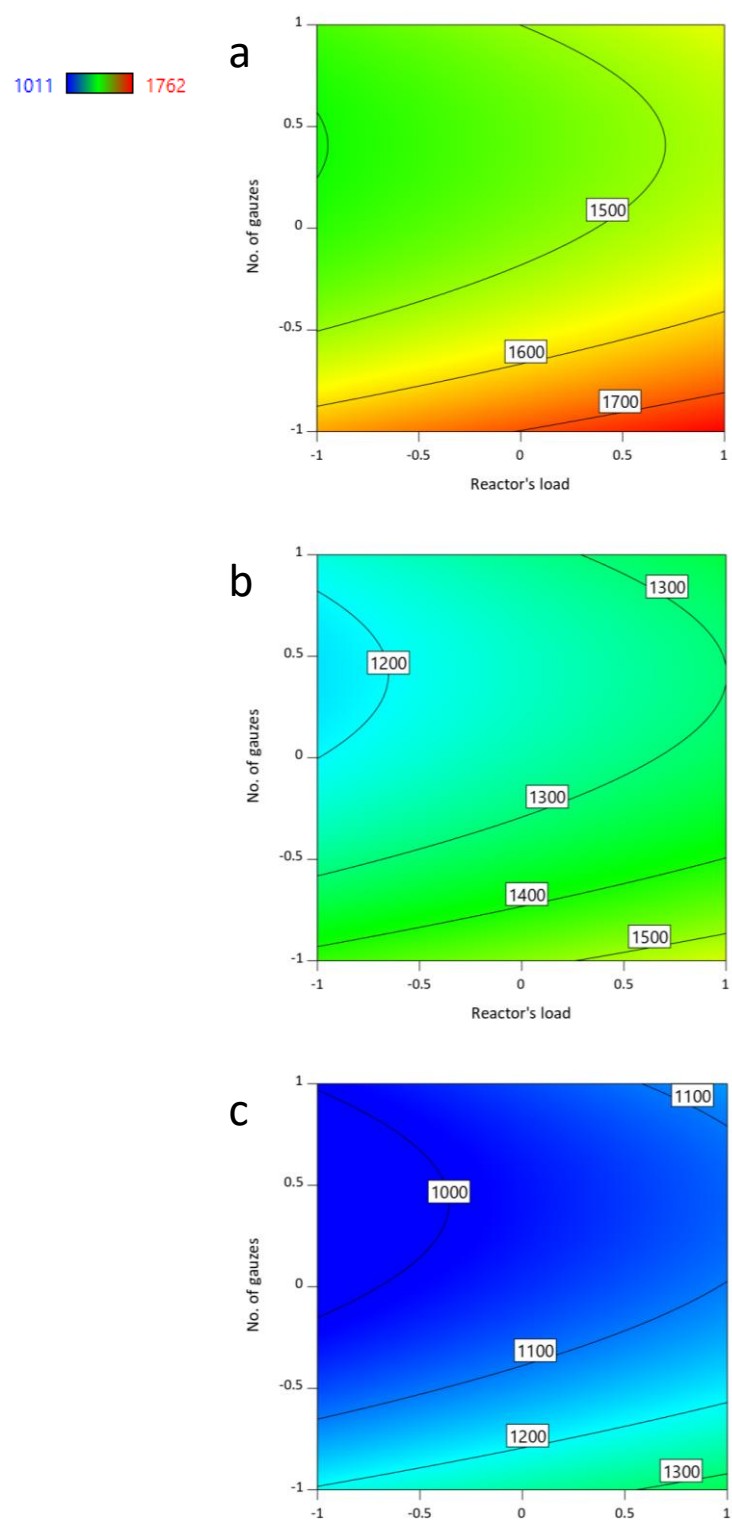

**Figure 6.** Interaction effect between the reactor's load ($X_1$) and the number of catalytic gauzes ($X_3$) at fixed temperature ($X_2$) on N$_2$O concentration in nitrous gases ($R_2$) by contour plot. (**a**) $X_2 = -1$; (**b**) $X_2 = 0$; (**c**) $X_2 = 1$.

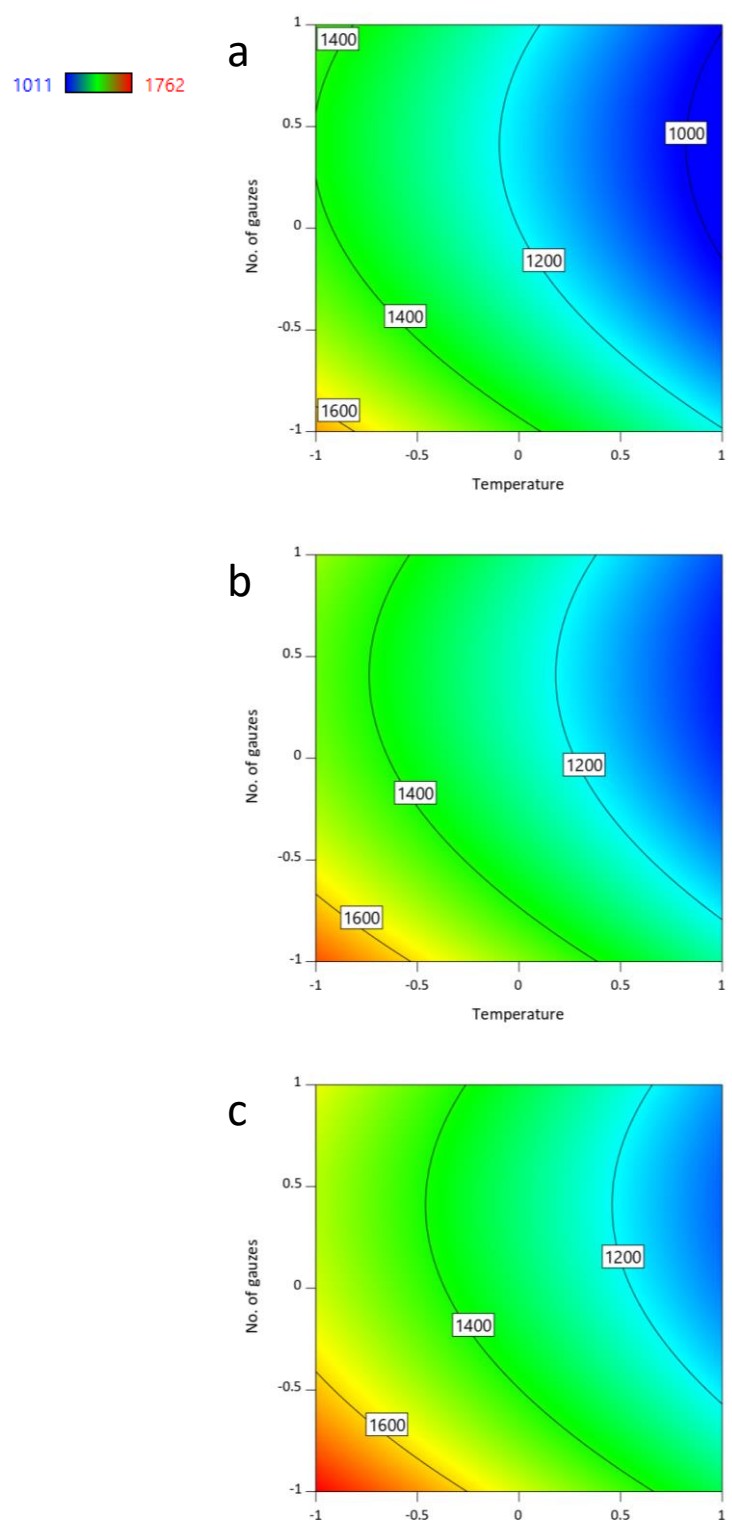

**Figure 7.** Interaction effect between the temperature ($X_2$) and the number of catalytic gauzes ($X_3$) at fixed reactor's load ($X_1$) on $N_2O$ concentration in nitrous gases ($R_2$) by contour plot. (**a**) $X_1 = -1$; (**b**) $X_1 = 0$; (**c**) $X_1 = 1$.

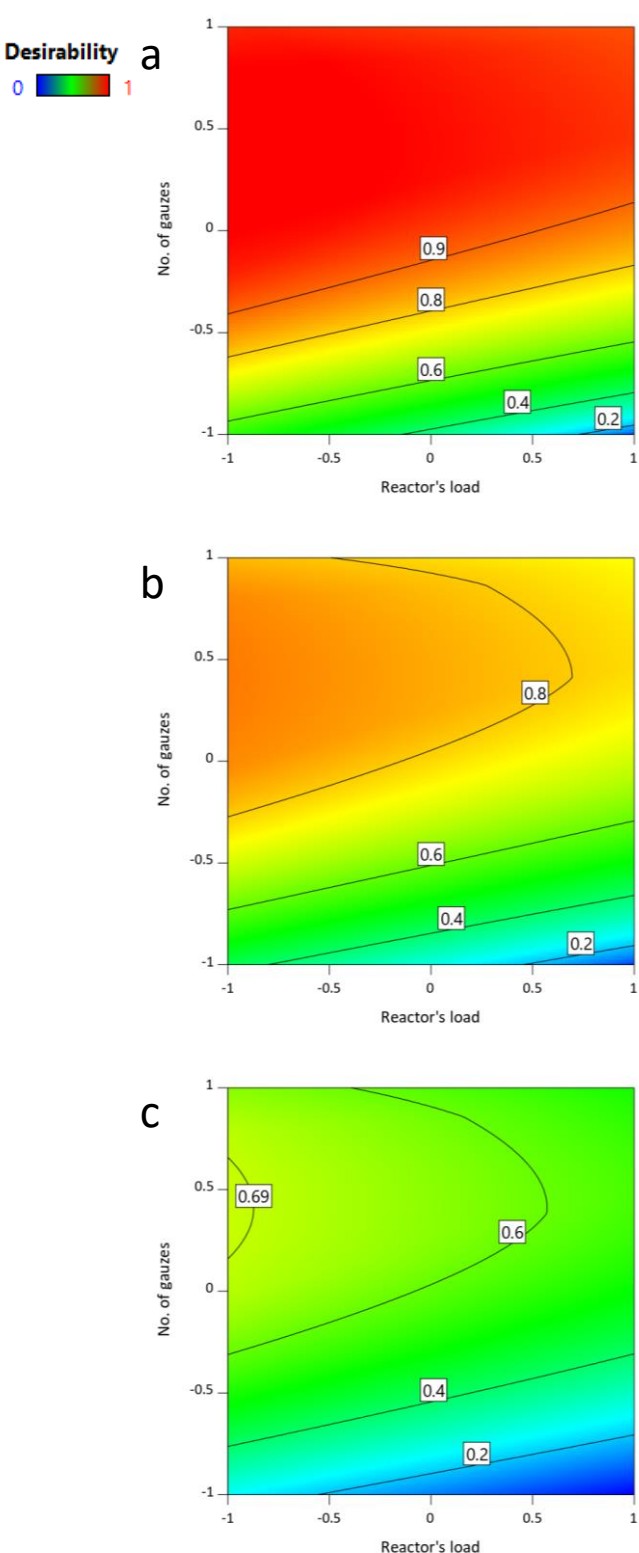

**Figure 8.** Desirability function plots. Effect of the reactor's load ($X_1$) and the number of catalytic gauzes ($X_3$) at three levels of temperature (**a**) $X_2 = -1$; (**b**) $X_2 = 0$; (**c**) $X_2 = 1$.

High values of desirability function ($DF > 0.9$) at 910 °C are described with dependency according to which for the load of 456 kg $NH_3/(m^2h)$, the sufficient number of catalytic gauzes is 8. However, for the maximum load studied, 10 catalytic gauzes should be applied. At the temperature of 910 °C and when all optimization criteria are met, the expected value of $N_2O$ concentration ranges from

1000 ppm to 1100 ppm (Figure 8a). Lowering the reaction temperature to 890 °C means that desirability function $DF > 0.8$ is within the region where the minimum catalyst gauzes is 9 for the loading not higher than 480 kg $NH_3/(m^2h)$ and 12 gauzes for load of 645 kg $NH_3/(m^2h)$ (Figure 8b). At this temperature, the expected concentration of $N_2O$ in nitrous gases ranges from 1180 to 1200 ppm. At the lowest temperature (within the studied range) of 870 °C, the highest value of desirability function is 0.69. For 12 gauzes and the load of 456 kg/$(m^2h)$, the expected concentration of $N_2O$ in nitrous gases is 1400 ppm.

Taking into account the amount of the primary emissions of $N_2O$ (the environmental aspect), it is favourable to conduct the reaction at the temperature of 910 °C. However, this leads to the increased platinum losses. Platinum losses at 910 °C are higher by approx. 25% as compared to losses at 890 °C and by 45% as compared to losses at 870 °C [1,2]. Lowering the reaction temperature to 890 °C with maintaining the optimal range of other parameters causes the increase of $N_2O$ concentration in nitrous gases by 100–200 ppm.

The assumption of other values of 'weight' and 'importance' for particular variables leads to obtain other profiles of desirability function. Under industrial conditions, the assumed value of 'weight' and 'importance' should take into account the process economics with regard to platinum losses.

### 2.6. Validation

Validation of the developed optimization model should be carried out under conditions specified as optimal. Optimization results indicate a wide set of parameters for which desirability function achieved high values. Therefore, in order to carry out additional measurements, the point with the independent variables value of: 1, 1, 1 was selected. This point is in the range of high desirability function value. In Table 8 levels of independent variables, results of validation experiment and predicted mean values of response variables with standard deviation are presented.

**Table 8.** The assumed levels of independent variables in validation studies and predicted responses values.

| Independent Variable | Reactor's Load | Temperature | Number of Gauzes |
|---|---|---|---|
| Level | 1 | 1 | 1 |
| Two-sided Confidence = 95%; Population = 99% | | | |
| Response variable | Experimental Data | Predicted Mean Value | Std Dev |
| $R_1$ | 96.2 | 96.3625 ± 0.4672 | 0.265216 |
| $R_2$ | 1120 | 1125.12 ± 83.37 | 52.9151 |

For the assumed independent variables, the values of predicted mean with 95% two-sided confidence intervals met by 99% of population were estimated based on the achieved mathematical model. High conformity of results expected according to mathematical models with the obtained measurements results was achieved.

## 3. Materials and Methods

### 3.1. Materials

Standard knitted gauzes made of platinum alloy with the addition of 10% wt. Rh made of 0.076 mm wire and specific weight of 600 g/$m^2$ were used for ammonia oxidation studies. The catchment gauzes which are most commonly used in industrial process were not used in studies. The prefiltered compressed air and gaseous ammonia were used as raw materials for ammonia oxidation reaction.

### 3.2. Experimental Procedure

Ammonia oxidation studies were performed in a pilot plant equipped with a flow reactor (inner diameter: 100 mm). The Pt-Rh catalytic gauzes were installed inside the basket. After initiating the reaction, the stable ammonia and air ratio was maintained in reaction mixture amounting to approx. 10.9% vol. The air–ammonia mixture temperature was controlled in such a manner as to obtain the temperature of nitrous gas as assumed in the experiment plan. Air–ammonia mixture temperature was variable in the range of 135–195 °C. The flow of the air-ammonia mixture was also controlled in order to obtain the assumed reactor's load. All experiments were conducted under the pressure of 0.5 MPa. The range of temperature of reaction at which studies were carried out is similar to that applied in industrial practice. The reactor's load was selected in such a manner as to ensure that the gas flow through the catalytic package in the range applied for medium-pressure industrial reactor namely 1–3 m/s. The scheme of the pilot plant is presented in Figure 9. For measuring ammonia oxidation efficiency, samples of ammonia-air mixture were taken at the inlet to the reactor and samples of nitrous gases were taken at the outlet of the reactor. For determination of $N_2O$ concentration in nitrous gases, only samples of nitrous gases were taken.

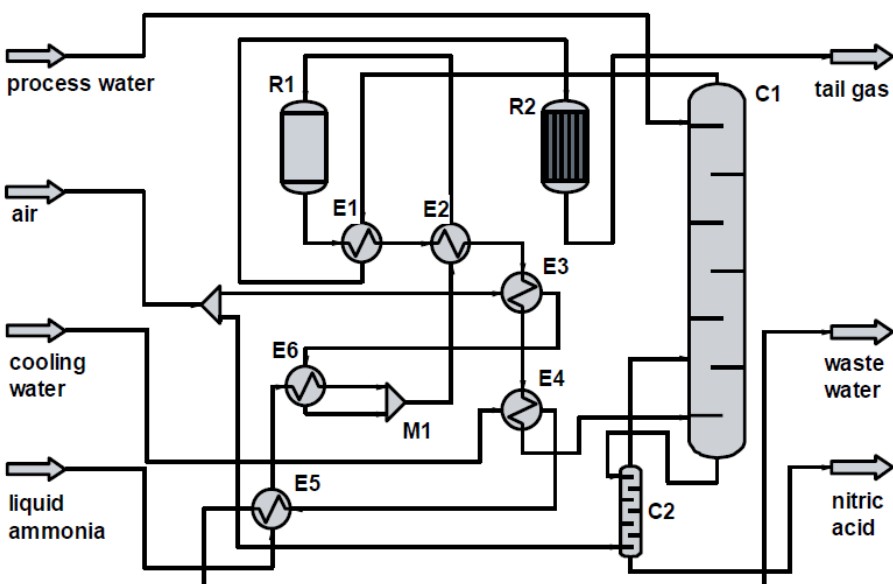

**Figure 9.** Scheme of the pilot plant. Symbols: C1—absorption column, C2—bleaching column, E1–E6—heat exchangers, M1—air–ammonia mixer, R1—ammonia oxidation reactor, R2—selective catalytic reduction reactor.

### 3.3. Analytical Methods

The ammonia oxidation efficiency was calculated based on concentrations of ammonia in the air-ammonia mixture at the inlet and concentrations of NO in nitrous gases at the outlet of the reactor. Both analysis were determined according to a titration method. Ammonia from air-ammonia mixture samples was absorbed in water with the formation of ammonia-water solution which was then titrated with sulphuric acid. The nitrous gases samples were absorbed in 3% water solution of hydrogen peroxide. After ensuring the sufficient period of time, NO oxidized completely to $NO_2$ and then, it reacted with water to $HNO_3$. The formed $HNO_3$ was titrated with the sodium hydroxide solution in the presence of an indicator.

Ammonia oxidation efficiency ($R_1$) was calculated according to the following formula:

$$R_1 = \left(\frac{C_2}{C_1}\right) \cdot 100\% \tag{3}$$

where: $C_1$—ammonia concentration in ammonia-air mixture, % $w/w$; $C_2$—concentration of oxidized ammonia, % $w/w$.

The result of each measurement is an average value, calculated from 7 independent samplings. The difference in the extreme individual values were not greater than $\pm 0.3\%$ in comparison to the average one.

$N_2O$ concentration in nitrous gases ($R_2$) was determined by gas chromatography using a Unicam 610 system with a discharge ionization detector. Gaseous samples were collected in the vacuum flasks containing 3% water solution of hydrogen peroxide. After the absorption of nitrous gases and water vapor condensation, exhaust gas from the flasks was injected to the gas chromatograph through 1 mL sample loop. The result of each measurement was an average value calculated from 3 independent samplings. The difference in the extreme individual values was not greater than $\pm 35$ ppm in comparison to the average one.

### 3.4. Statistical Methods

The experimental procedure was carried out according to Box–Behnken design matrix. The reactor's load ($X_1$), the temperature of nitrous gas specifying the temperature of reaction ($X_2$) and the number of catalytic gauzes ($X_2$) were selected as independent variables. Ammonia oxidation efficiency ($R_1$) and $N_2O$ concentration in nitrous gas ($R_2$) were specified as response variables. Each level of independent variables were coded according to the Equation (4).

$$X_i = \frac{x_i - x_0}{\Delta x_i} \tag{4}$$

where, $X_i$ is the dimensionless, coded level of independent variable ($-1$, 0 or 1), $x_i$ is the actual value of the independent variable, $x_0$ is the value of the independent variable at the centre point, $\Delta x_i$ is the step change in $x_i$.

Ranges and levels of independent variables are presented in Table 9.

**Table 9.** Coded and uncoded levels of independent variables used in experiments.

| Independent Variable: | $X_1$ | $X_2$ | $X_3$ |
|---|---|---|---|
| | Reactor's Load, kg $NH_3/(m^2h)$ | Temperature of Reaction, °C | Number of Catalytic Gauzes, pcs |
| Coded variable level: | | | |
| Low level ($-1$) | 456 | 870 | 5 |
| Mid-level (0) | 582 | 890 | 10 |
| High level ($+1$) | 708 | 910 | 15 |

The total number of the experiments ($N$) was calculated using the Equation (5).

$$N = 2k(k - 1) + c_0 \tag{5}$$

where, $k$ is the number of independent variables, $c_0$—number of the replicates run of the centre point (in our research $c_0 = 3$). For three independent variables, the total number of experiments assumed in the plan was 15.

The experiments were conducted in a randomized order to avoid the influence of uncontrolled variables on the dependent responses.

A mathematical relationship between the independent variables and response variables was determined by fitting the experimental data with second-order polynomial Equation (6).

$$R_i = b_0 + \sum_{i=1}^{3} b_i X_i + \sum_{i=1}^{3} \sum_{j=1, \, i<j}^{3} b_{ij} X_i X_j + \sum_{i=1}^{3} b_{ii} X_i^2 \tag{6}$$

where $R_i$ is the estimate response variable, $b_0$, $b_i$, $b_{ii}$, $b_{ij}$ are regression coefficients fitted from the experimental data, $X_i$, $X_j$ are coded independent variables, listed in Table 9.

The significance of the model equation, individual parameters, were evaluated through ANOVA with the confidence interval (CI) of 95%. A simultaneous optimization of several dependent variables requires the application of multi-criteria methodology. In this case, the desirability function (*DF*) was used. The particular desirability functions are combined using the geometric mean which allows to achieve overall desirability function [16], according to Equation (7).

$$DF = \left( (d_1)^{w_1} \times (d_2)^{w_2} \times \ldots x\ (d_n)^{w_n} \right)^{1/\sum w_i} \tag{7}$$

where $n$ is the number of responses, $d_i$ is an individual response desirability, $w_i$ is a response 'weight'.

The adjustment of the shape of particular desirability function can be performed by assigning the specified 'weight.' Setting a different 'importance' for each objective with respect to the remaining objectives is also possible. For these studies, identical 'weight' for all the independent variables and response variables was assumed. Desirability function assigns values from 0 to 1 where 1 means meeting all the optimization criteria. It is not always necessary to search for the solution aiming at achievement of the highest value of desirability function but it is vital to search for the set of parameters which would meet the optimization objectives to the particular extent (e.g., $DF > 0.75$). The statistical software used to experimental design and analysis was Design Expert 11.0.6.0 version (Stat-Ease, Inc., Minneapolis, MN, USA).

## 4. Conclusions

The conducted studies allowed us to develop statistically significant mathematical models describing the course of variables of ammonia oxidation efficiency and $N_2O$ concentration in nitrous gases depending on three selected independent variables.

The design of the experiment allowed the reduction of the costs of studies and to achieve a number of results accurate for modelling. It was found that, within the studied range of variability, the temperature of reaction has no significant effect statistically on the achieved ammonia oxidation efficiency, whereas it has the effect on the amount of $N_2O$ formed in the side reaction (primary emission of $N_2O$).

The developed models were used to optimize the process. As a result of this optimization, the set of the independent variables was developed for which optimization assumptions are met, which are expressed as a high value of desirability functions. It is possible to specify the optimum number of gauzes with the determined reactor's load for the studied package of catalytic gauzes.

In validation experiments, the developed model of desirability function achieved the high conformity of experimental values with the expected ones.

The presented methodology can be used to minimize the primary $N_2O$ emission at high ammonia oxidation efficiency. It can be applied for optimization of operating parameters of ammonia oxidation reactor with two types of catalysts: catalytic gauzes and catalyst for high temperature of $N_2O$ decomposition. As a result, it is possible to obtain the set of independent variables ensuring low $N_2O$ emission and to meet the binding environmental regulations.

**Author Contributions:** Conceptualization, M.I.; Formal analysis, M.I. and A.D.-I.; Investigation, M.I. and J.R.; Methodology, M.I.; Supervision, M.W.; Visualization, A.D.-I.; Writing—original draft, M.I.

**Funding:** This research received no external funding.

**Conflicts of Interest:** The authors declare no conflict of interest.

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
