# Peer review of "Optimization of Ammonia Oxidation Using Response Surface Methodology"

_catalysts, doi:10.3390/catal9030249_

Round 1
Reviewer 1 Report
This manuscript needs improvement before it is accepted for publication.
The comments are given below:
in section 2.3, Analytical methods (page 3)
please give the detailed description of analytical methods of NH3 and NO
Legends description in Fig.1 & Fig.2 were missing, it is not clear for readers what the dots with different colors in figure 1 or figure 2 represent for .
· In page 18, Table 9, line 274, 95% CI low ,
what CI means for ?
in Table 8, please translate Polish “Maksymalizacja produkcji” into English.
Author Response
Authors are grateful for all reviewer's opinions. The manuscript layout was amended according to the Catalysts layout requirements. Affiliation was completed. The corrected changes was presented as "Track Changes" in word file and additionally new manuscript version without visible corrections as PDF. Introduction section was extended for a more in-depth understanding background of the study . References section was updated.In fig. 4-8, 3D plots (on the left-hand side) were removed because they show identical data as contour plots (on the right-hand side) and the scale of colors was added in this place. Tables and drawings were formatted. Numbers of figures and tables were changes after changing the layout of the manuscript. Answers to reviewer’s comments are attached (marked in red).

Reviewer 2 Report
This manuscript reports a tentative study for optimization of well-known ammonia oxidation by surface response methodology.
The work severely fails due to the following reasons. First, it is well-known that ammonia oxidation is under external mass transfer limitation. For this reason reaction temperature poorly affects ammonia conversion; for the same reason, reaction kinetics is linear with respect to ammonia concentration in the gas phase and the kinetic constant is related to the mass transfer coefficient. This also means (second reason) that NH3 conversion exponentially depends on contact time (i.e. on the catalyst weight); so, the quadratic dependence on X3 found by the authors is conceptually wrong and could be considered an approximation. This approximation could be (obviously) valid only within the investigated experimental range; as a matter of fact, if extended to X3 values higher than 1, conversion would decrease by increasing contact time, which is not the case. So (third reason), the declared optimization could be performed only within the experimental conditions used for this work, suggesting that N2O formation is always higher than the limiting value, and thus resulting in no interest for engineering purposes.
In conclusion, I suggest to reject the manuscript.
Author Response
Authors are grateful for all reviewer,s opinions. These are valuable comments for correcting this manuscript and will be taken into account for publishing the other results in the future. The manuscript layout was amended according to the Catalysts layout requirements. Affiliation was completed. The corrected changes was presented as "Track Changes" in word file and additionally new manuscript version without visible corrections as PDF. Introduction section was extended for a more in-depth understanding background of the study . References section was updated. The remaining sections were re-organized pursuant to suggestions. In fig. 4-8, 3D plots (on the left-hand side) were removed because they show identical data as contour plots (on the right-hand side) and the scale of colors was added in this place. Tables and drawings were formatted. Numbers of figures and tables were changes after changing the layout of the manuscript. Answers to reviewer’s comments are attached (marked in red).

Reviewer 3 Report
In this manuscript, the authors describes the optimization of ammonia oxidation using response surface methodology. While the article is generally well written, these are lots of issues that need to be addressed. This is as follows:
1) In line 29, the word should be semi-organic products.....
2) In line 32, the sentence should be: 'the catalytic oxidation of ammonia to nitrogen oxide (NO) with the use of.....' The current sentence as written does not make sense.
3) In line 36, the word 'of ' should be removed. The sentence should be ...'Among numerous catalyst[2-8].....'
4) In line 48, what do you mean by: 'At the end of the 80' of the 20th century?' 80' ?
5) In line 104, the sentence should be : The statistical software used in experimental design.....
6) The placement of the plots in figures 5, 6 and 7 are very distracting and can be very confusing to say the least. It would be very helpful to readers if each of these plots are placed on a separate page instead of spreading each of them between two pages.
7) In lines 290, 291, 293...etc, there should be a space between the number and the degree celcius. Example: 910 oC not 910oC.
8) There are many other instances of grammatical errors that need to be corrected to avoid any linguistic ambiguity.
In summary, this manuscript has the potential to benefit its targeted audience if each of the above concerns are adequately addressed.
Author Response
Authors are grateful for all reviewer's opinions. These are valuable comments for correcting this manuscript and will be taken into account for publishing the other results in the future. The manuscript layout was amended according to the Catalysts layout requirements. Affiliation was completed. The corrected changes was presented as "Track Changes" in word file and additionally new manuscript version without visible corrections as PDF. Introduction section was extended for a more in-depth understanding background of the study . References section was updated. In fig. 4-8, 3D plots (on the left-hand side) were removed because they show identical data as contour plots (on the right-hand side) and the scale of colors was added in this place. Tables and drawings were formatted. Numbers of figures and tables were changes after changing the layout of the manuscript. Answers to reviewer’s comments are attached (marked in red).

Reviewer 4 Report
A model for optimizing ammonia oxidation has been developed by considering reactor load, temperature and the number of catalytic gauzes as decisive parameters and describing their influence on ammonia oxidation efficiency and N2O concentration.
In general, technical quality of the manuscript is low due to a series of small inaccuracies, e.g. misleading labelling of variables or fragmentary description of some issues, and comprehensibility is therefore restricted. Furthermore, the paper seems to be incomplete since Table 9 in Chapter 3.5 needs to be discussed and thus, this chapter should be extended.
The following topics should also be considered:
L 29: … aromatic …
L 32: … to nitrogen oxide (NO) …
L 33: … and absorption …
L 50: … because of its medical applications …
Chapter 2: The experimental part is very brief. A reference and/or figure of the plant would be helpful.
L 85: A reference related to Box-Behnken design (Ref [19]?) should be given.
L 113: R1 and R2 are defined in Table 2. This is confusing and a definition should be given in the text.
L 125, Eq. (3) and the following sentence: Please check the labelling of variables (Ri, Rj?). Xj is already defined by Eq. (1).
Tables 4-7: Please use a regular footnote or describe “Significance” in the text. The insignificant entries should also be named in the tables.
Figures 1 and 2: Please specify colours of data points; Please correct “Residulas”; Please correct the figure captions, e.g. by using one caption which describes the four subfigures, and avoid inaccurate labelling such as “Predicted”.
Figure 3: Labelling of the figure on the left is cut.
Table 8: Please translate the entries in the second line.
L 258 and in the following: Please check the figure numbers (Fig. 9?).
L 292: … platinum losses …
Conclusion:
The conclusion should precisely reflect the study. Pt losses have not been in the focus of the work. Please specify “BAT requirements”.
Author Response

(The authors gave the same response as above.)

Round 2
Reviewer 2 Report
I carefully read the Authors' reply and the revived manuscript. Unfortunately, in my opinion the main drawbacks I raised have not been overcome. So, I still suggest rejection
Author Response
We do not agree with your opinion
Reviewer 4 Report
A series of changes has been made so that quality of the manuscript improved.
Please check the numbering of chapters, it is not correct.
Figs. 2 and 3: Labelling of the subfigures is missing.
Author Response
The text was re-checked and minor errors were corrected.
Please check the numbering of chapters, it is not correct.
It was corrected
Figs. 2 and 3: Labelling of the subfigures is missing.
It was corrected